# Studies of Buried Layers and Interfaces of Tungsten Carbide Coatings on the MWCNT Surface by XPS and NEXAFS Spectroscopy

**Danil Sivkov** [1,2,*], **Sergey Nekipelov** [2], **Olga Petrova** [2], **Alexander Vinogradov** [1],
**Alena Mingaleva** [2], **Sergey Isaenko** [2], **Pavel Makarov** [2], **Anatoly Ob'edkov** [3], **Boris Kaverin** [3],
**Sergey Gusev** [4], **Ilya Vilkov** [3], **Artemiy Aborkin** [5] and **Viktor Sivkov** [2]

1   Faculty of Physics, Federal State Budgetary Educational Institution of Higher Education "Saint-Petersburg State University", 199034 St. Petersburg, Russia; asvinograd@yahoo.de
2   Institute of Physics and Mathematics, Komi Science Center Ural Division of the Russian Academy of Sciences, 167982 Syktyvkar, Russia; nekipelovsv@mail.ru (S.N.); teiou@mail.ru (O.P.); amingaleva@gmail.com (A.M.); isaenko@geo.komisc.ru (S.I.); mkrvpa@gmail.ru (P.M.); sivkovvn@mail.ru (V.S.)
3   Hybrid Nanomaterials Laboratory, G.A.Razuvaev Institute of Organometallic Chemistry of the Russian Academy of Sciences, 603950 Nizhny Novgorod, Russia; amo@iomc.ras.ru (A.O.); kaverin@iomc.ras.ru (B.K.); mr.vilkof@yandex.ru (I.V.)
4   Department of Magnetic Nanostructures, Institute for Physics of Microstructures of the Russian Academy of Sciences, 603950 Nizhny Novgorod, Russia; gusev@ipmras.ru
5   Institute of Engineering, Vladimir State University Named after Alexander and Nikolay Stoletovs, Gorky Str. 87, 600000 Vladimir, Russia; aborkin@vlsu.ru
*   Correspondence: d.sivkov@spbu.ru or danjorno@yandex.ru; Tel.: +7-821-2428-4352

**Abstract:** Currently, X-ray photoelectron spectroscopy (XPS) is widely used to characterize the nanostructured material surface. The ability to determine the atom distribution and chemical state with depth without the sample destruction is important for studying the internal structure of the coating layer several nanometers thick, and makes XPS the preferable tool for the non-destructive testing of nanostructured systems. In this work, ultra-soft X-ray spectroscopy methods are used to study hidden layers and interfaces of pyrolytic tungsten carbide nanoscale coatings on the multi-walled carbon nanotube (MWCNT) surfaces. XPS measurements were performed using laboratory spectrometers with sample charge compensation, and Near Edge X-ray Absorption Fine Structure (NEXAFS) studies using the Russian–German dipole beamline (RGBL) synchrotron radiation at BESSY-II. The studied samples were tested by scanning and transmission electron microscopy, X-ray diffractometry, Raman scattering and NEXAFS spectroscopy. It was shown that the interface between MWCNT and the pyrolytic coating of tungsten carbide has a three-layer structure: (i) an interface layer consisting of the outer graphene layer carbon atoms, forming bonds with oxygen atoms from the oxides adsorbed on the MWCNT surface, and tungsten atoms from the coating layer; (ii) a non-stoichiometric tungsten carbide $WC_{1-x}$ nanoscale particles layer; (iii) a 3.3 nm thick non-stoichiometric tungsten oxide $WO_{3-x}$ layer on the $WC_{1-x}$/MWCNT nanocomposite outer surface, formed in air. The tungsten carbide nanosized particle's adhesion to the nanotube outer surface is ensured by the formation of a chemical bond between the carbon atoms from the MWCNT upper layer and the tungsten atoms from the coating layer.

**Keywords:** XPS; NEXAFS; TEY; MWCNT; $WC_{1-x}$/MWCNTs; MOCVD

## 1. Introduction

X-ray photoelectron spectroscopy (XPS) is an analytical technique that has been used for many years to study surfaces. The formation, development and application of the XPS method are described in detail in the widely known books by Kai Siegbahn [1,2] and in many other monographs and reviews published later [3–10] and articles published over the past decade [11–13]. In modern conditions of rapid spectral equipment development and the emergence of high-intensity synchrotron radiation sources, XPS methods have been actively used to study nanostructured systems and materials. The electron exit-depth energy dependence data [14] make it possible to determine the atom distribution and their chemical state variation with depth without sample destruction. This allows us to study the inner structure of coating layers several nanometers thick and makes XPS the preferred non-destructive testing method for interfaces and complex nanosized coatings. The development of the systems of the effective sample charge compensation during the XPS measurement significantly expanded the studied object's range and allowed us to study the finely dispersed materials. In this work, XPS is used to study hidden layers and interfaces of pyrolytic tungsten carbide coatings on the multi-walled carbon nanotube (MWCNT) surface.

MWCNTs are modern nanostructured materials with high chemical and heat resistance, conductivity, hardness and strength. In combination with a large surface, this makes MWCNTs a promising material for usage in various applications. Moreover, the MWCNT surface modification with inorganic and organic coatings allows us to change the properties of resulting composites. With a coating thicknesses of about several nanometers, the composites acquire unique physicochemical properties [15]. It is known that carbon nanotubes have properties similar to graphite, in particular, metals poorly wet their surface [16]. This means that without MWCNT surface treatment, it will be difficult to achieve good adhesion of metal-containing particles and coatings. Hybrid materials based on MWCNTs are the subject of intensive investigations [17] because of a large choice of the methods of their synthesis (electrochemical techniques, various deposition methods such as atomic layer deposition, pulsed laser deposition, physical and chemical vapor deposition, etc.). The MWCNT wall's modification with various metal-containing nanoparticles [18,19] can expand the range of the nanotubes' functional properties [20], giving them the required magnetic [21], catalytic [22] and electronic [23] properties. The $Cu_2O$/Cu/MWCNT nanocomposite has successfully proved to be a catalyst for the germanium tetrachloride reduction with hydrogen [24,25]. The MWCNT surface decoration with iron nanoparticles was used for nanoelectronic device manufacture, data storage in magnetic media and in magnetic resonance imaging [26].

Nanostructured tungsten carbide has received significant attention because of its enhanced tribomechanical properties [27] as well as fuel cell application [28–30]. In chemical catalysis, tungsten carbide (WC) can be used as a catalyst in hydrogenation, dehydrogenation, isomerization and synthesis of hydrocarbon materials. For example, recently, Kim et al. [31] reported that the hydrogenation ability of WC catalyst can be improved by synthesizing nano-sized WC. The use of the nano-WC catalyst for the hydrocracking of vacuum residue caused the lowest coke formation and the highest yield of commercial liquid fuel products in comparison with commercially available bulk WC catalyst.

Several researchers have demonstrated that a combination of nanostructured tungsten carbide with graphitic carbon—for example, carbon nanotubes—has found wide application in the synthesis of nanocomposites due to their superior physical and chemical properties. MWCNT-based hybrid materials with the surface decorated by tungsten carbides nanoparticles or thin films form a new class of functional carbon nanomaterials. Using MWCNTs allows us to increase the specific surface area of the deposited nanoparticles due to the stabilization of highly dispersing tungsten carbide nanoparticles.

Tungsten carbide nanocoatings are also of interest primarily in the context of the fabrication of high-efficiency electrocatalysts for fuel cells, where it should replace platinum, which is very costly [32]. A considerable number of studies that focus on the synthesis of hybrid materials with MWCNT and tungsten carbide nanocoatings and on their possible applications in various research fields have already been published. Keller et al. [33] obtained one-dimensional tungsten carbide nanostructured

material. Carbon nanotubes were used as a carbonaceous template and carbon source for carburizing pre-impregnated tungsten oxide into tungsten carbide under synthesis conditions such as 1300 °C for 7 h, resulting in one-dimensional metal-free tungsten carbide. Shi et al. [34] prepared multi-walled carbon nanotube–tungsten carbide composites by the reduction and carbonization process using MWCNTs and $WO_3$ precursor by molecular-level mixing and calcinations at 800 °C for 20 h with additional carbon source $CH_4$.

Recently, carbon nanotubes have attracted increasing attention as an effective catalyst support for tungsten carbide catalyst due to their unique properties such as high current carrying ability, thermal conductivity, mechanical strength and chemical stability [35–38]. Li et al. [35] observed that WC/CNTs nanocomposite prepared by the approach of surface decoration and in situ reduction–carbonization show that the electrocatalytic activity of the sample is higher than that of purified carbon nanotubes, hollow globe tungsten carbide with mesoporosity and granular tungsten carbide. These results indicate that (tungsten carbide)/(carbon nanotubes) composite creation is the effective method of increasing the tungsten carbide electrocatalytic activity. The Pt supported on WC-modified MWCNT catalysts (PtWC/MWCNTs) were synthesized by a combination of organic colloidal and intermittent microwave heating methods [36]. The results proved the better performance of the PtWC/MWCNT catalyst than that of the Pt/C for methanol oxidation. Rahsepar et al. [37] grew MWCNTs directly on the graphite rod surface by using the CVD process. Tungsten carbide was used as a platinum co-catalyst to modify the resulting MWCNTs with subsequent deposition of Pt catalyst nanoparticles using the electrochemical pulse deposition method. According to the electrochemical measurements results, the prepared catalysts show better electrocatalytic performance towards methanol oxidation compared to a commercial Pt/C catalyst with about four-times higher Pt loading. Wang et al. [38] reported that they prepared a Ni-WC/MWCNT catalyst for the electrooxidation of urea in alkaline conditions. The micro-morphology and composition of the hybrid materials are determined by transmission and scanning electron microscopy (TEM and SEM), and X-ray diffraction (XRD) analysis. The characterization results indicate that the Ni nanoparticles are uniformly distributed on the WC/MWCNT framework, and the Ni-WC/MWCNT catalyst shows an improved activity for the urea electrooxidation that is higher than those of Ni-WC/C and Ni/C catalysts. The authors believe that the higher activity on the Ni-WC/MWCNT catalyst is attributed to the support effect of MWCNT as well as the synergetic effect between Ni and WC. Herein, we report a simple method for synthesis of $WC_{1-x}$/MWCNT composites by metal–organic chemical vapor deposition with tungsten hexacarbonyl as a precursor [39]. It was found that the initial mass ratio of MWCNTs and $W(CO)_6$ was dependent on the desired thickness of the $WC_{1-x}$ coating. It was found that if the initial MWCNT/$W(CO)_6$ ratios were less than three, the coating was discontinuous. The size of the $WC_{1-x}$ particles varied from 10 to 50 nm. The hybrid materials synthesized with a larger initial amount of $W(CO)_6$ had continuous tungsten carbide coatings. On the basis of AA5049 aluminum alloy reinforced by MWCNTs and MWCNTs coated with $WC_{1-x}$ nanoparticles using high-energy ball milling and high-temperature consolidation, the two types of bulk nanocomposites were successfully produced [40]. The study of the synthesized nanocomposites mechanical properties demonstrated the composites strengthened by MWCNTs coated with $WC_{1-x}$ nanoparticles had higher microhardness and elastic modulus compared to the original aluminum alloy AA5049.

However, to date, despite the active study of the WC/MWCNT composite, there are still a lot of unresolved issues related to both the nanocomposite-producing technology and the diagnostics of its structural and physicochemical properties. These include, in particular, the mechanism of coating atoms adhesion to the nanotube outer graphene surface, tungsten carbides and oxides non-stoichiometry, the atomic composition of an interface nanotube–(coating layer), the coating layer structure, and more.

To solve the issues, the WC/MWCNT nanocomposites were studied by TEM, SEM, Raman scattering, XRD, ultra-soft X-ray spectroscopy (Near Edge X-ray Absorption Fine Structure, NEXAFS) and XPS. Particular attention was paid to the study of XPS spectra in the region of C1, O1 and W4f ionization edges. This is due to the fact that XPS makes it possible to study changes in the composition

of near-surface layers with depth in the nanoscale and to test changes in the atomic and chemical composition of the surface and coating without destruction and modification [41,42]. Moreover, XPS studies determine the chemical composition and effective thickness of the MWCNT surface coating and the mechanism of adhesion to the nanotube surface. Since the WC/MWCNT nanocomposite samples are finely dispersed powders that are charged during XPS measurements, it is necessary to use a photoelectron spectrometer with a sample charge compensation system during the experiment. XPS studies were carried out at the resource center "Physical methods of surface investigation" (Saint Petersburg University Research park) by Thermo Fisher Scientific ESCALAB 250Xi X-ray spectrometer. NEXAFS studies were performed using synchrotron radiation (SR) of the Russian–German dipole beamline (RGBL) at BESSY-II.

## 2. Materials and Methods

### 2.1. Materials

MWCNTs were synthesized in the Hybride Nanomaterials Laboratory of G.A. Razuvaev Institute of Organometallic Chemistry of the Russian Academy of Sciences (RAS) via metal–organic chemical vapor deposition (MOCVD) in a tubular quartz reactor. General characteristics of MWCNTs used in the work: average outer diameter—80 nm, average length—300 μm, carbon purity—(97–98) wt%, Fe-based catalyst residue—(5.5–6.0) wt%.

Materials list: especially pure toluene (produced by JSC ECOS, Moskow, Russia); ferrocene $(C_{10}H_{10})Fe$ (98%, produced by Aldrich, St. Louis, MO, USA); tungsten hexacarbonyl $W(CO)_6$ (97%, produced by Aldrich, St. Louis, MO, USA); gaseous argon (produced by LLC SPE "Salyut-gas", Nizhny Novgorod, Russia, TU 2114-011-106-818-63-2005). The volume fraction of argon is not less than 99.994%.

### 2.2. Experimental Details

#### 2.2.1. Synthesis of MWCNTs

For the synthesis of MWCNT, the MOCVD method via the pyrolysis of ferrocene and toluene mixture in an argon flow (flow rate 500 cm$^3$/min) at atmospheric pressure in a tubular quartz reactor was used. The synthesis procedure is described in detail in [43].

The schematic illustration of the experimental setup used for the synthesis of radial-oriented aligned MWCNTs is presented in Supplementary Figure S1. The MWCNT array preparation was carried out on the inner surface of a cylindrical quartz insert at 825 °C under the argon flow rate of 500 sccm and ferrocene sublimation temperature at 130 °C. Using this method, we obtained nanotubes containing residual iron nanoparticles inside the channels of nanotubes (Fe-filled MWCNTs). The obtained MWCNTs arrays were mechanically removed from the quartz insert in the form of radially oriented layers and then grinded in a rotary-type disperser with a knife rotation speed of ~400 rpm.

#### 2.2.2. Synthesis of (Pyrolytic WC)/MWCNT Nanocomposites

For the deposition of the pyrolytic tungsten carbide coatings on the MWCNT surface, MWCNTs weighed of 0.4 g and tungsten hexacarbonyl $W(CO)_6$ (weighed from 0.4 to 3.2 g) was used. The pyrolytic tungsten carbide deposition on the MWCNT surface was carried out on the experimental setup, which scheme is showed in Supplementary Figure S2, in accordance with the procedure outlined in [38].

The tungsten hexacarbonyl decomposition proceeded according to the reaction:

$$MWCNTs + W(CO)_6 \xrightarrow{300\ ^\circ C} (Pyrolytic\ WC)/MWCNTs + gaseous\ products. \tag{1}$$

The prepared nanocomposite (pyrolytic WC)/MWCNTs was stored in a volume filled with high-purity argon.

## 2.3. Characterization

MWCNTs and hybrid materials synthesized for the current study were characterized using a number of chemical and physical analysis methods. XRD, TEM, SEM and the energy-dispersive X-ray spectroscopy (EDS) were conducted using equipment of Center "Physics and technology of micro- and nanostructures" at The Institute for Physics of Microstructures of RAS, and Raman Spectroscopy—using equipment of the Centre of Collective Use at the Institute of Geology of Komi Science Centre of the Ural Branch of RAS. All samples were studied by NEXAFS spectroscopy using synchrotron radiation from RGBL at BESSY II Storage Ring (Berlin, Germany) [44,45]. XPS studies were carried out using the equipment of the resource center of the Science Park of St. Petersburg State University "Physical methods for surface research". The features of experimental studies and the characteristics of used equipment are described in more detail in [42].

Raman studies were conducted at room temperature using an Ar laser with the power of 1 mW and wavelength of 488 nm. Spectra were recorded by the grating of 600 g/mm in the range 100–4000 $cm^{-1}$. The spatial and spectral resolution was 1 μm and 1 $cm^{-1}$, respectively. Each spectrum was the result of three accumulations with a 10 s exposure. The spectra flitting were performed by LabSpec 5.36 software.

All NEXAFS spectroscopy data were obtained at the Berliner Elektronenspeicherring für Synchrotronstrahlung (BESSY) using radiation from the Russian–German beamline (RGBL) [44,45]. All spectra were obtained using total electron yield (TEY) mode. For energy calibration the energy positions of the first- and second-order light-excited Au $4f_{7/2}$ lines were used. The flux in arbitrary units was measured by the Au plate. The resolution of photon energy was less than 0.05 eV.

XPS studies were carried out using AlKα radiation (1486.6 eV). The C1, O1, W4f spectra were measured at the pass energy of 50 eV and the survey spectra at 100 eV. The studies were performed at room temperature in vacuum of $10^{-10}$ mbar with using the system of an electron-ion charge compensation of the samples

All Raman, NEXAFS, and XPS measurements were performed for a series of samples at different points for each sample, and their results had good repeatability.

## 3. Results and Discussion

The samples of the initial MWCNTs and the (pyrolytic WC)/MWCNT nanocomposites prepared on their basis were tested by TEM, SEM, XRD and Raman spectroscopy.

### 3.1. Initial MWCNT Research

Figure 1 shows cross-sectional SEM images of the radially-oriented MWCNTs (a) and MWCNTs (b,c) synthesized from ferrocene and toluene obtained under selected conditions after grinding in a rotary-type disperser. Nanotubes are close-packed in an array radially-oriented to the substrate surface—the inner surface of the quartz cylinder.

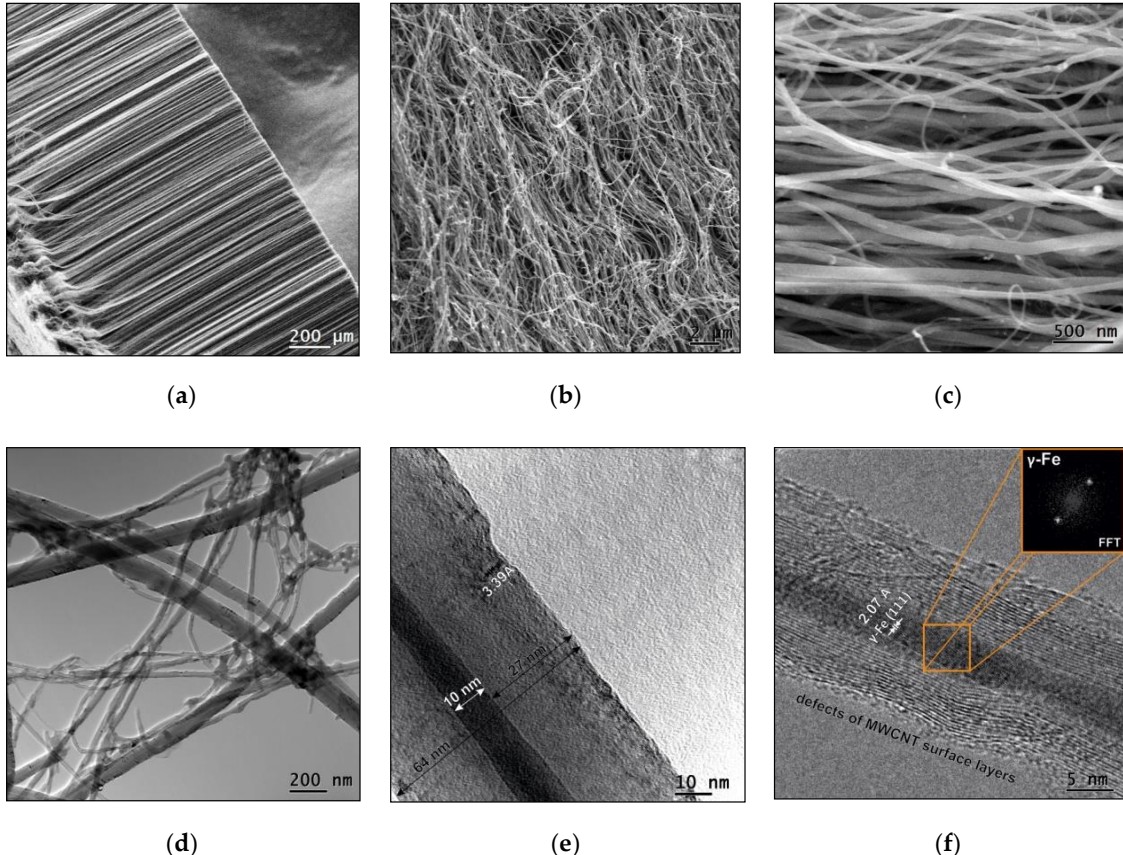

**Figure 1.** Cross-sectional scanning electron microscopy (SEM) images of the radially-oriented aligned multi-walled carbon nanotubes (MWCNTs) (**a**), the initial MWCNTs after grinding in a rotary-type disperser (**b**,**c**). The transmission electron microscopy (TEM) image of MWCNTs (**d**) and high-resolution transmission electron microscopy (HRTEM) images of an iron catalyst ($\gamma$-Fe)-filled nanotube (**e**,**f**). The MWCNT dimensions and carbon interplanar spacing 3.39 Å are marked in (**e**). The (**f**) image shows the fast Fourier transform insert, that indicates the crystallinity of the filling ($\gamma$-Fe interplanar spacing 0.207 nm).

The SEM studies revealed the presence of MWCNTs with different diameters (Figure 1c). According to TEM data (Figure 1c), the average outer diameter of the synthesized carbon nanotubes is about 80 nm and their lengths ranging from hundreds of micrometers to several millimeters. According to high-resolution transmission electron microscopy (HRTEM) image, the MWCNT lateral surface is formed by layers of graphene with average distances between them equal to 0.339 nm (Figure 1e). Defects in the form of residual graphene layers on the MWCNT outer surface are shown in Figure 1e,f. The MWCNT internal channel diameter is, on average, 10 nm. HRTEM images also show the presence of nanotube partial filling with $\gamma$-Fe nanoparticles (Figure 1e,f). The crystalline iron particle fast Fourier transform (FFT) is shown in the insert (Figure 1f). According to the EDS results, the atomic composition of the obtained MWCNTs after transfer into air mainly consists of carbon (95%) and oxygen (5%). This indicates the presence of a small amount of oxides on the nanotube surface.

The initial MWCNT X-ray powder diffraction analysis (Figure 2a) demonstrates that the main features correspond to the graphite peaks (002), (100), (101) and (004). Moreover, diffraction peaks (111), (200) prove the residual iron catalyst ($\gamma$-Fe) presence.

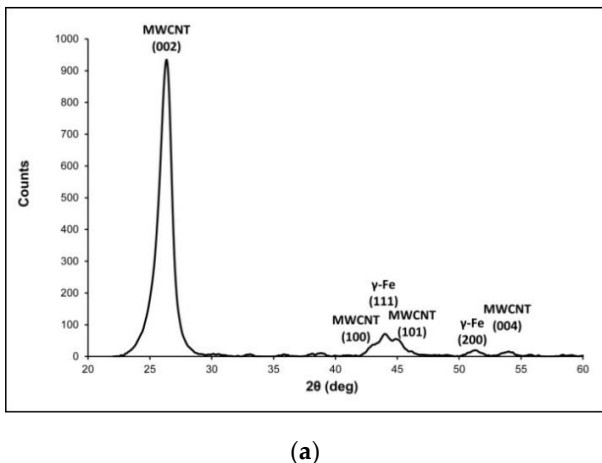
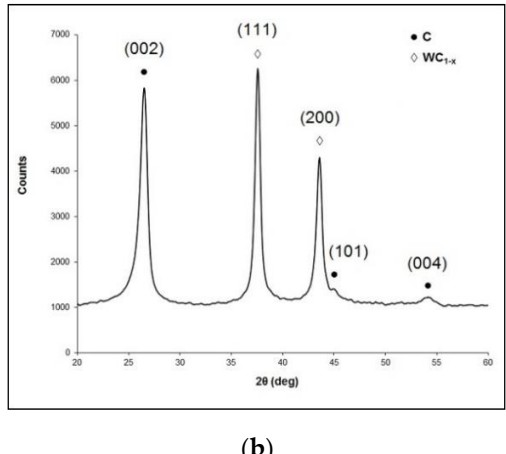

| (a) | (b) |

**Figure 2.** The X-ray powder diffraction patterns of (**a**) the initial MWCNTs and (**b**) as-prepared composite material sample MWCNTs coated with pyrolytic tungsten carbide with initial MWCNTs/W(CO)$_6$ ratio of 1:3 (sample 1).

### 3.2. (Pyrolytic WC)/MWCNT Nanocomposite Research

During the study it was found that the deposition conditions and the mass of pyrolytic tungsten carbide coating highly determine the hybrid nanomaterials morphology. In Figure 3 the SEM images of hybrid materials obtained with initial MWCNT/W(CO)$_6$ ratios of 1:3 (sample 1), 1:4 (sample 2) and 1:5 (sample 3) are shown. The coating weight gain was 0.3, 0.426 and 0.723 g for samples 1–3 respectively. It was discovered that if the ratio of MWCNTs/W(CO)$_6$ is smaller 1:3 then the coating is discontinuous (Figure 3a). In addition, nanoparticle sizes vary from few nm to few tens nm. The hybrid materials synthesized with an initial ratio of 1:4 have a continuous coating of pyrolytic tungsten carbide, but at the same time there are MWCNT areas free of coating. According to TEM data, the MWCNTs coated with pyrolytic tungsten carbide (Figure 3b) have an average thickness of about 20–30 nm. The hybrid materials with a large initial ratio of 1:5 have a continuous coating of pyrolytic tungsten carbide with a thickness of more than 100 nm (Figure 3c). It is shown by SEM (Figure 3b,c) that the pyrolytic tungsten carbide coating thickness varies widely, and the coating is granular. All this suggests that coatings are not deposited as continuous thin layers covering the entire MWCNT surface, they become continuous only if grains increase sizes and coalesce at a relatively high initial W(CO)$_6$ concentration. According to the EDS results (Figure 3d), the atomic composition of the obtained MWCNTs coating is mainly represented by tungsten, carbon and oxygen.

The EDS spectra were recorded for various points of the sample. The error of the atom concentrations determined from them was 30%. This is due to the sample characteristics, namely the cylindrical surface of the structural elements, the presence of spaces between them and the coating multilayer composition. For the samples with a thin coating, less than the electron penetration depth, the contribution to the intensity of the characteristic MWCNT carbon K$_\alpha$-line is large, and the coating quantitative chemical analysis cannot be carried out. Samples with a thick coating are preferable in this regard, but the contribution of the MWCNT CK$_\alpha$-line introduces a distortion into the quantitative analysis. In addition, the emission of carbon and oxygen atoms can be absorbed by a tungsten compound coating, which will also increase the error in determining the concentrations of these atoms in the sample. As a result of EDS measurements, the atomic composition of the sample with a thick coating (sample 3) of pyrolytic tungsten carbide after its transfer into air estimated as ~15% (W), ~10% (O) and ~75% (C). In the case of samples 1 and 2 with a thinner and non-continuous coating, the atomic composition according to EDS varies within 5–10% for tungsten, 5–10% for oxygen and 90–80% for carbon. The presence of oxygen atoms in the samples can be associated with the tungsten oxide formation on the outer surface of the pyrolytic tungsten carbide coating.

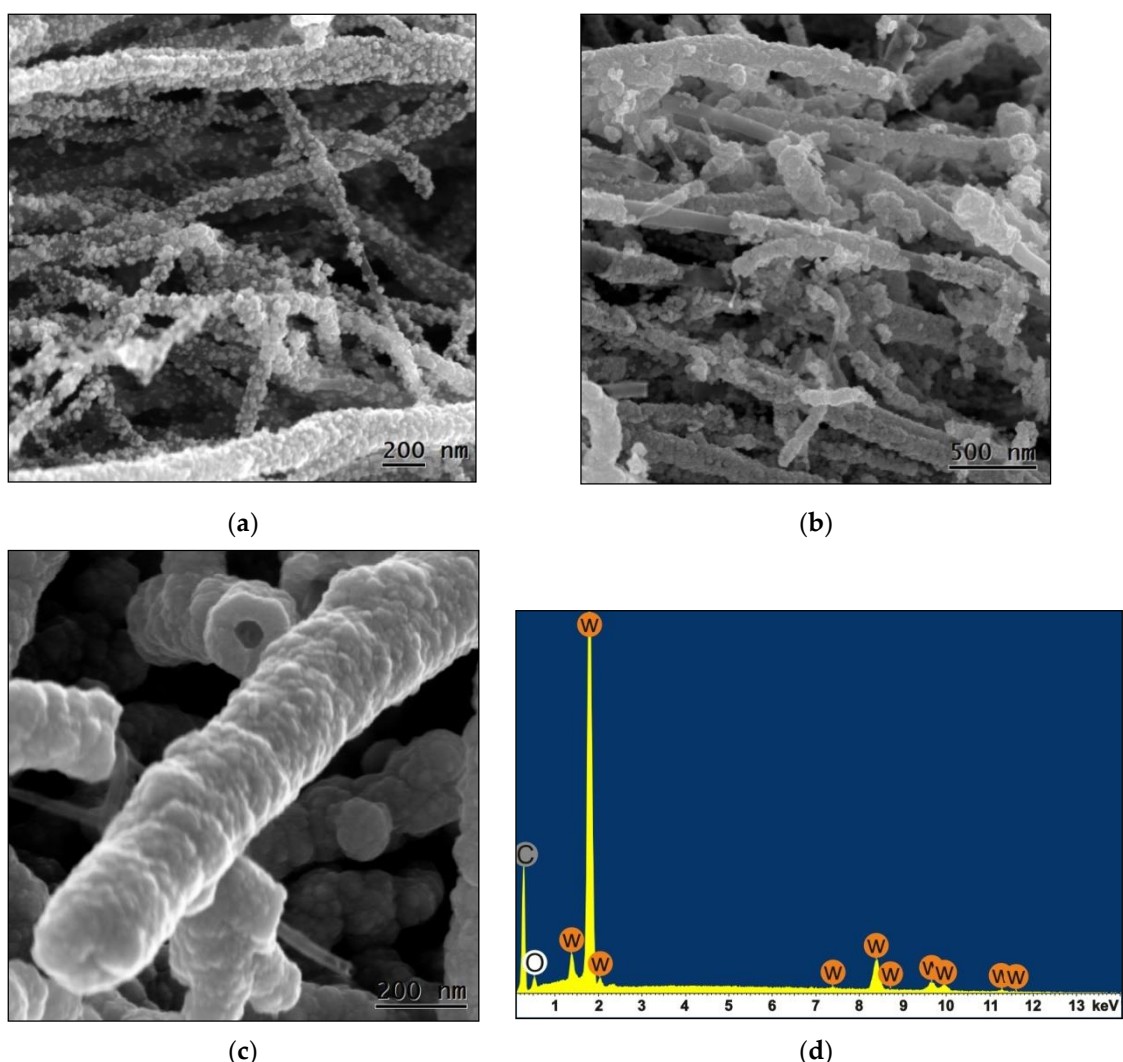

**Figure 3.** The SEM images of the composite material sample coated with pyrolytic tungsten carbide MWCNTs with initial mass ratios of precursors (MWCNTs:W(CO)$_6$) of 1:3 (sample 1) (**a**), 1:4 (sample 2) (**b**) and 1:5 (sample 3) (**c**). (**d**)—the decrypted EDS profile obtained from the MWCNT coating shown in (**c**).

To determine the phase composition of the coating, the method of X-ray diffractometry was used. A typical X-ray powder diffraction pattern of the as-prepared composite material sample MWCNTs coated with pyrolytic tungsten carbide with the initial MWCNT and W(CO)$_6$ precursor ratio of 1:3 (sample 1) is presented in Figure 2b. The XRD results show the presence of the carbon nanotubes phase and the nonstoichiometric WC$_{1-x}$ phase. It should be noted that the diffraction pattern of the sample does not contain reflections related to tungsten oxides. This suggests that the oxide content in the pyrolytic tungsten carbide coating is negligible.

The pyrolytic tungsten carbide coating on the surface of MWCNT was identified as an ultra-disperse WC$_{1-x}$ Fm3m phase with lattice parameter a = 4.14 Å (fcc structure; JCPDS card no. 20-1316) with no indication of WC or W$_2$C. Earlier [46] formation of this tungsten carbide phase was identified where mixtures of MWCNTs and W(CO)$_6$ were subjected to ultrasonic treatment in argon flow at T = 95 °C for 3 h in hexadecane. The nonstoichiometric WC$_{1-x}$ phase is likely stable only in the case of nanosized crystallites.

The MWCNTs coated with WC$_{1-x}$ nanoparticles TEM results are given in Figure 4. The TEM image of WC$_{1-x}$/MWCNTs in Figure 4a,b shows that WC$_{1-x}$ in initial stages formed aggregates of small nanoparticles. Furthermore, with the continuation of the deposition process, these aggregates

subsequently form islands of coatings on the MWCNT surface and then continuous coatings. Figure 4b insert shows the HRTEM image of a cluster nanoparticle. The blue color indicates the system of clusters interplanar distances in a reflective position. Crystal clusters are disoriented within a single nanoparticle as evidenced by the FFT shown on Figure 4c.

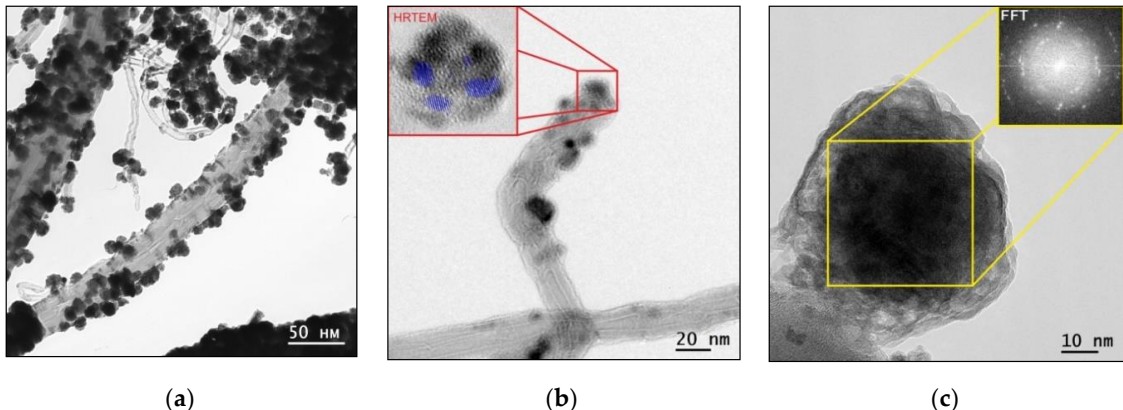

**Figure 4.** The TEM images of (**a**) WC$_{1-x}$/MWCNTs, (**b**) WC$_{1-x}$/MWCNT with the HRTEM insert, and (**c**) the HRTEM image of WC$_{1-x}$ nanoparticles with the fast Fourier transform (FFT) insert (sample 1).

The MWCNTs coated with WC$_{1-x}$ nanoparticles TEM results show that their interior is entirely composed of tungsten carbide. This allows us to conclude that tungsten oxides are formed not during pyrolysis, but as a result of contact with atmospheric oxygen and form a thin oxide layer on the tungsten carbide nanoparticles surface.

### 3.3. Raman Spectra of the Nanocomposites and Initial MWCNTs

In order to obtain information about the coatings and to test the structure of nanotubes, the samples were analyzed by Raman spectroscopy. The Raman scattering spectrum was studied by using 488 nm laser light. The corresponding experimental curves are shown in Figure 5. In the MWCNT spectrum, in addition to the main peak G (1586 cm$^{-1}$), peak D (1361 cm$^{-1}$), characteristic of non-crystalline materials, appears. The degree of sample graphitization can be determined through the $I_D/I_G$ (peak intensities ratio). This value depends on number of impurities and defects in the composition of MWCNT graphene layers. The lower the ratio, the higher the graphitization and the less the amount of impurities and defects. For the MWCNTs under study, the value $I_D/I_G = 0.45$, which indicates the high purity of the samples [47].

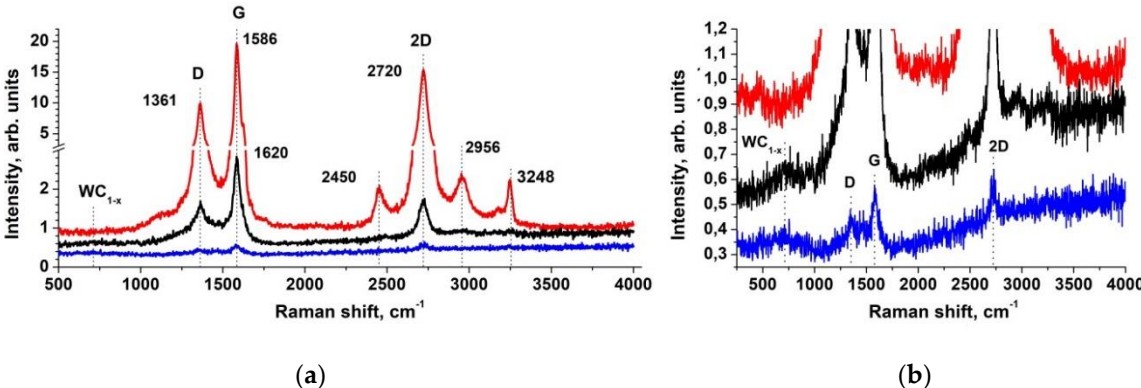

**Figure 5.** The Raman spectra in wide (**a**) and narrow (**b**) energy range of the initial MWCNTs (red), (pyrolytic tungsten)/MWCNT nanocomposite sample 2 (black) and sample 3 (blue).

Figure 5 shows the Raman spectra of the initial MWCNTs and nanocomposites (samples 2 and 3). The WC/MWCNT spectra in Figure 5 contain all the characteristic peaks of the initial nanotube. Since their ratios $I_D/I_G$ are close to 0.5, one can make the conclusion that the metal deposition does not destroy the nanotube structure. The intensities of D and G peaks in the Raman spectrum are greatly reduced for sample 2 and become very small for sample 3 compared with those for the initial MWCNTs. These noticeable changes are due to the strong absorption of the incident, scattered by the nanotube laser radiation by a pyrolytic tungsten carbide coating layer and the uncoated nanotube surface area decrease. The penetration depth of the exciting laser light into the WC and $WC_{1-x}$ coating could roughly be estimated to a 30–80 nm [48–50]. In doing so, the penetration depth into the amorphous and crystalline $WO_3$ films is very high, because all the films are transparent above λ = 400 nm [51]. Therefore, the strong suppression of the MWCNT surface Raman modes is caused only in the pyrolytic tungsten carbide layer. From previously published works [52–62], it is known that crystalline $WO_3$ corresponds to the Raman peaks at 275, 880 and 945 $cm^{-1}$. However, this structure is not observed in the Raman spectra of the studied samples. It can be seen from Figure 5b that in the region 600–1000 $cm^{-1}$, some broad peaks are visible and can be created by tungsten carbide or oxide phases. The strong broadening of this band suggests that the formed compounds present a high structural disorder, indicating low crystallinity [60–62]. The presence of the band in the Raman spectra is characteristic of $WC_{1-x}$ nanoscale crystallites and its appearance in the spectra of samples 2 and 3 is probably associated with a nanocomposite structure formed by $WC_{1-x}$ nanocrystals [55,60,63,64], which is consistent with diffraction analysis.

*3.4. NEXAFS Spectroscopy Research of the Nanocomposites and Initial MWCNTs*

The MWCNTs and $WC_{1-x}$/MWCNT nanocomposites were studied using NEXAFS spectroscopy method in TEY mode. The background radiation contributions in the incident beam and in the TEY signal was determined using the method detailed in [42,65,66].

To normalize the TEY signal from MWCNTs and MWCNT-based nanocomposites, the use of gold TEY signals divided by the absorption cross section of the Au atom is necessary

In the work [42], it was shown that to normalize the TEY signal from MWCNTs and MWCNT-based nanocomposites, the use of the TEY signal of gold divided by the Au atom absorption cross section is necessary. The normalized TEY signal can be obtained as [42]:

$$\sigma_s(E_0) \sim Y_s\ (E_0)/Y_{Au}\ (E_0), \tag{2}$$

where $Y_s\ (E_0)$ and $Y_{Au}\ (E_0)$ is the TEY signal of sample and Au-plate.

Figure 6 shows the spectral dependences in the same relative units (left scale) of the TEY signal for the MWCNTs and nanocomposite $WC_{1-x}$/MWCNTs (Sample 2) together with the atomic absorption cross sections spectral dependences in the absolute scale in Mb (right scale) obtained by modeling the samples atomic composition. It is known that the absorption cross section of any compounds far from their constituent atoms absorption edges can be represented with good accuracy as the sum of the atomic absorption cross sections with weight coefficients equal to the fraction of their atomic composition [67]. This allows us to estimate the corresponding atom's percentage by comparing the calculated and experimental absorption cross sections dependences outside the absorption edges. From EDS data, it is known that the MWCNT contains mainly carbon and several percentage-points of oxygen atoms, and the nanocomposite contains tungsten, carbon and oxygen. In the case of the initial MWCNTs, good agreement with the experimental curve is observed at weights of 97% and 3% for carbon and oxygen atoms, respectively, which correlates with the EDS data. In the case of the $WC_{1-x}$/MWCNT nanocomposite (Sample 2), the optimal atomic ratios are 10%, 80% and 10% for tungsten, carbon and oxygen, respectively. It should be noted that in the case of sample 2, the coating of the nanotube surface with tungsten carbide nanoparticles is incomplete and nonuniform in thickness. For this reason, the contribution to the TEY signal is made in addition to the $WC_{1-x}$ covering layer

and tungsten oxides thin layers, the MWCNT free surface and the surface of MWCNTs coated with $WC_{1-x}$ layers of different thicknesses. Therefore, the proportions of carbon and oxygen atoms are qualitative characteristics, in contrast to tungsten atoms, which are contained only in the pyrolytic tungsten carbide coating layer. This is also due to the fact that the absorption cross sections of tungsten atoms in the energy range 250–800 eV outside the NEXAFS C1 and O1 absorption edges exceed the cross sections of carbon and oxygen atoms by an order of magnitude or more [67], and therefore make the main contribution to the nanocomposite normalized TEY signal.

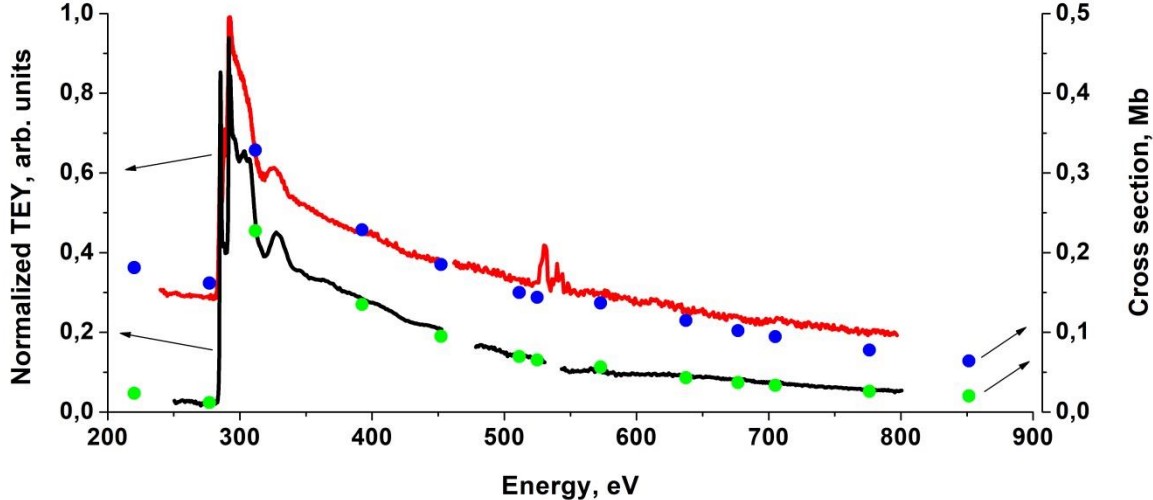

**Figure 6.** The normalized total electron yield (TEY) signal of MWCNTs (black) and nanocomposite Sample 2 (red) in arbitrary units (**left scale**) and atomic cross sections sums [67] C(97%) + O(3%) (green), W(10%) + C(80%) + O(10%) (blue) in Megabarns (**right scale**).

As noted earlier in the analysis of SEM data, the $WC_{1-x}$ coating has good adhesion to the outer surface of the MWCNTs. This indicates that the pyrolytic tungsten carbide layer and the MWCNT nouter graphene layers chemically interact with each other. To analyze this interaction, the C1 NEXAFS spectra of the initial MWCNTs and composites were compared. The π*- and σ*-resonances, characteristic of the initial MWCNT spectrum, are retained in the composite spectrum (Figure 7a). It should be noted that the MWCNT C1 spectrum has a very low-intensity broad band in the 286–289 eV region, which is probably associated with the interaction of carbon with a small amount of oxides adsorbed on the nanotube outer surface. The presence of the oxide is confirmed by XPS studies. The unchanged structure characteristic of the MWCNT C1 spectrum indicates that in the composite, the MWCNT outer layers remain almost intact. However, in the intermediate energies region (285.4–291.8 eV) between π*- and σ*-resonances, there is an extra structure in the form of explicit peak C together with low-intensity peaks A, B, E and shoulder D with energies of 288.5 eV, 287.2 eV, 287.5 eV, 289.2 and 290.4 eV, respectively.

In addition, it can be seen from the partial C1 dependences of the normalized TEY signals (Figure 7b) that the integral values of the partial C1 absorption cross sections (the area under the spectral dependences) noticeably decrease upon transition from MWCNTs to nanocomposites. There is also a strong decrease in the intensity of π*-resonance by more than a factor of two in the partial NEXAFS C1 absorption spectrum of the nanocomposite as compared with the spectrum of the initial MWCNTs. In this case, the absorption decrease in the continuum in the region of 320 eV is 1.2. The latter is in good agreement with the ratio of the carbon atom content in the MWCNTs and $WC_{1-x}$/MWCNT composite (sample 2) obtained from the EDS analysis (95% and 75%) and from the atomic cross-section model sum (Figure 6) equal to 97% and 80%.

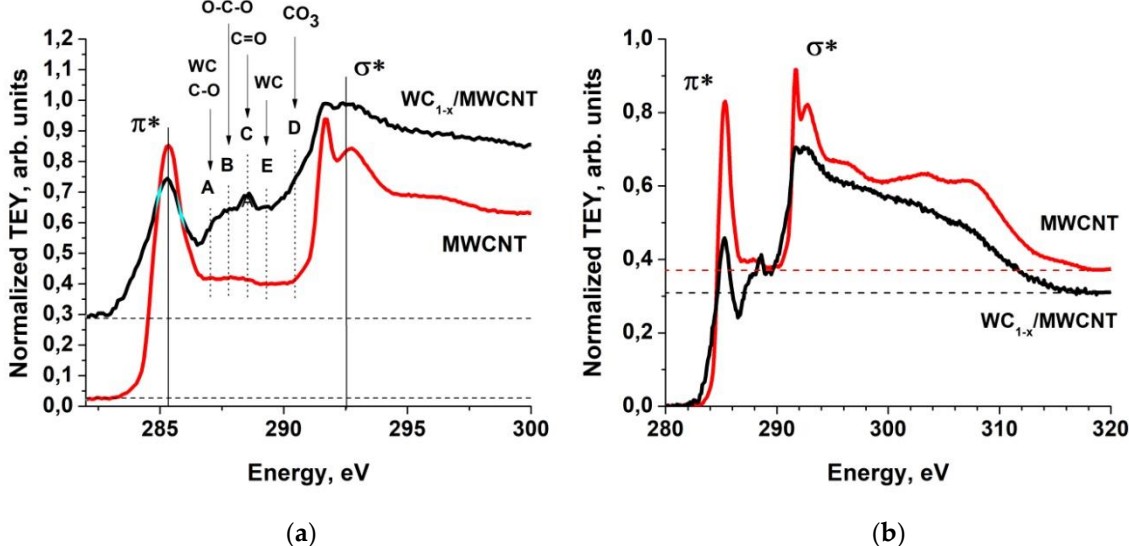

**Figure 7.** Normalized TEY signal of MWCNTs (red) and nanocomposite sample 2 (black) in arbitrary units in near the Near Edge X-ray Absorption Fine Structure (NEXAFS) C1 absorption edge (**a**). The arrows indicate the fine-structure elements energy positions of the carbon oxygen's [68,69] and WC [70–73] near the NEXAFS C1 edge. Dash-dotted lines indicate the partial C1 absorption cross sections isolation. Partial C1s normalized TEY signal of MWCNTs (red) and nanocomposite sample 2 (black) in arbitrary units (**b**). The dashed lines indicate the absorption levels in the continuum.

The absorption decrease in the NEXAFS region is due to the weakening of the photoelectrons, secondary and Auger electrons flow from the outer surface of the MWCNT in the layer of pyrolytic tungsten carbide and the MWCNT free-surface reduction. However, this decrease is partially compensated by the contribution to the absorption from the $WC_{1-x}$ coating layer carbon atoms. Therefore, the NEXAFS C1 spectrum of the $WC_{1-x}$/MWCNT composite is a superposition of the spectra of the MWCNT surface, noncoated and coated with a thin $WC_{1-x}$ layer, the interface between the coating and the nanotube outer surface and the $WC_{1-x}$ layer of different thicknesses. The participation of some carbon atoms of the nanotube in the oxides and tungsten carbides formation also leads to a decrease in the π*-resonance intensity compared with the absorption intensity in the continuum. While the absorption outside the NEXAFS region in the continuum is determined by the carbon atoms concentration, which remains unchanged during the formation of the coating on the initial nanotube surface.

Unfortunately, the correct separation of the contributions to the NEXAFS C1 spectrum mentioned above is not possible due to the complexity of the spectrum decomposition. The peaks A–E energies are equal to the energies of the elements in the C1 NEXAFS spectrum of the graphite oxides, and correspond to electron transitions from the C1 level to π* unoccupied orbital of C–O–C (A), C–O (B), C=O (C) [68] and $[CO_3]^{2-}$ (D) [69] atomic groups and to hybridization of C2p and metal W5d orbitals 286.8 eV (A) and 289.0 eV (E) of WC [70–72].

In Figure 7a, the positions of the peaks are indicated by arrows. The presence of A–D peaks in the composite NEXAFS C1 spectrum indicates the carbon oxides and carbides formation in the sample. This can occur both on the MWCNT and $WC_{1-x}$ surfaces, and inside the coating layer. The tungsten carbide deposition during the tungsten pentacarbonyl $W(CO)_6$ pyrolysis at a temperature of 300 °C is a complex chemical process. The main chemical reaction can be accompanied by a number of secondary reactions with the formation of free carbon and tungsten oxides of various compositions in the form of impurities in the final $WC_{1-x}$ coating. In this case, in the initial stage, the carbide and oxide formation can occur with the participation of the nanotube outer surface carbon atoms. However, the HRTEM studies showed (Figure 4) that tungsten carbide coatings have good adhesion to the MWCNT outer surface, and the $WC_{1-x}$ nanoparticles inner part consists entirely of tungsten

carbide. This suggests the presence of chemical bonds between the MWCNT carbon atoms, tungsten atoms from the coating layers, and oxygen atoms from the oxides on the nanotube surface. When the composite is transferred into the air, the tungsten oxide formation due to its contact with atmospheric oxygen should be expected.

Therefore, the good adhesion of $WC_{1-x}$ to MWCNTs can be explained by the formation of a chemical bond through tungsten of pyrolytic tungsten carbide coating and the MWCNT outer surface. However, in the initial stage of the W-metal atom deposition on the MWCNT surface at a temperature of 300 °C, carbon oxides can also be formed. Tungsten carbide and carbon oxide spectral study is impossible due to the superposition of their structures in the NEXAFS C1 spectra. However, carbon oxides, metal oxides and metal carbides separate studies can be performed using the XPS method.

### 3.5. XPS Research of the Nanocomposites and Initial MWCNTs

Figure 8 shows the XPS survey spectrum of the initial MWCNTs and $WC_{1-x}$/MWCNT nanocomposite (sample 2). Figure 9a–c shows their photoelectron spectra in the C1, O1 and W4f regions, respectively. The photoelectron spectra of the initial MWCNTs are in good agreement in the peaks number, their energy positions and half-widths with the nanotubes XPS spectra from [73–77]. The MWCNT C1 spectrum consists of an intense asymmetric peak A (284.5 eV) with a smooth tail from the short-wave side (its binding energy corresponds to one of C1's electrons in the nanotubes graphene layers) and a wide P band with an energy of 291.0 eV due to additional excitation of π plasmon. It is significant that in the MWCNT XPS C1 spectrum (Figure 9a) there are no separate peaks related to carbon oxides, but in the survey spectrum in Figure 8 there is a low-intensity O1 peak in the 532 eV region, which indicates the presence of oxides on the initial nanotube surface. This band, as can be seen in Figure 9b, consists of two peaks B (531.6 eV) and C (533.4 eV), which are associated with transitions in the OH⁻ [78] group and $H_2O$ [79] water molecule adsorbed on the nanotube surface upon their transfer to the atmosphere. The effect of these compounds on the states of carbon atoms on the nanotube outer surface is probably the reason for the appearance of a wide absorption band of 286–289 eV (Figure 7a) in the NEXAFS C1 absorption spectra of the initial MWCNTs. The percentage of oxygen atoms in the initial MWCNTs was determined by using the Thermo Avantage software package from comparing the integrated peak intensities in the C1 and O1 spectra of the MWCNT and equals to 2.3%. This result refines the data from EDS and NEXAFS spectroscopy. The initial MWCNT XPS spectra analysis demonstrates their high purity degree.

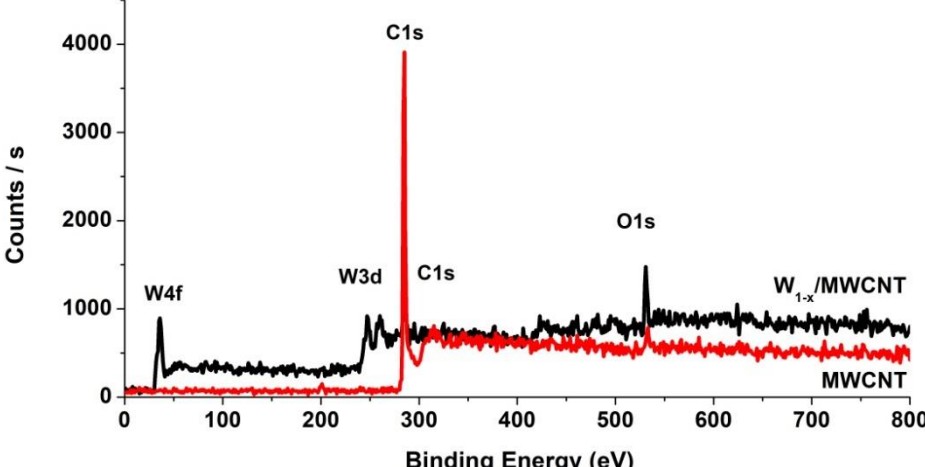

**Figure 8.** The X-ray photoelectron spectroscopy (XPS) spectra of the $WC_{1-x}$/MWCNTs sample 2 (black) and initial MWCNTs (red) in a wide energy range.

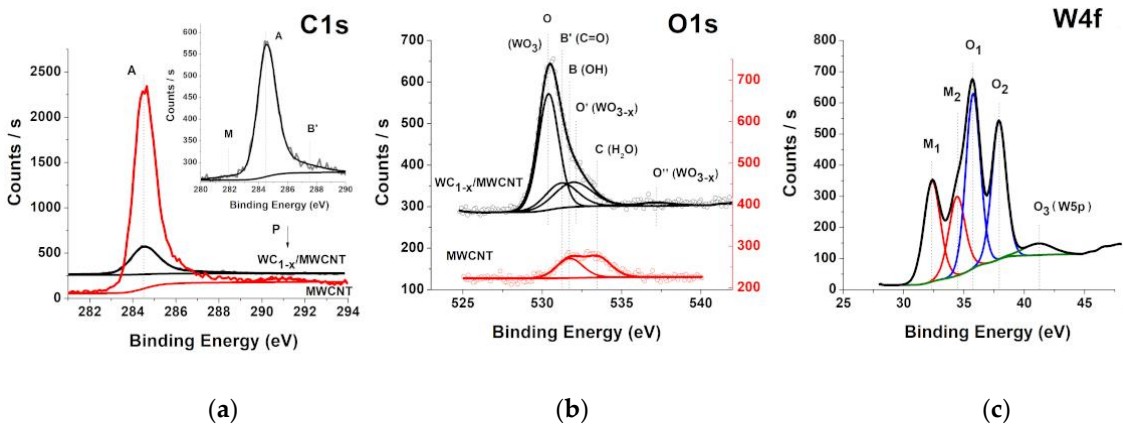

**Figure 9.** The C1 (**a**), O1 (**b**) XPS spectra of the WC$_{1-x}$/MWCNTs sample 2 (black) and initial MWCNTs (red). The inset in (**a**) shows the composite XPS C1 spectrum. (**c**) the W4f XPS spectra (black) fitting (colored) of the sample 2.

For the WC$_{1-x}$/MWCNT composite, the XPS signals of Tungsten 4f and 3d (centered at ~35 eV and ~245 eV), C1s (~285.0 eV) and O1s (~532 eV) were detected (Figure 8). A comparison of the XPS C1 spectra of the initial nanotubes and the WC$_{1-x}$/MWCNT composite (Figure 9a) shows a strong decrease in intensity and an increase in half-width by 0.33 eV of the composite main peak A (284.5 eV), corresponding to the bond between carbon atoms in graphene layers. Moreover, there are two additional low-intensity details M (282.9 eV) and B′ (287.5 eV) in the WC$_{1-x}$/MWCNT spectrum. The energy position of the peak M corresponds to the C1 binding energy in tungsten carbides [80–82]. Therefore, its appearance is associated with transitions in carbon atoms of the WC$_{1-x}$ coating layer. The weak peak energy B′ corresponds to the binding energy of the C1 electron in the C=O group [83,84], the presence of which in the nanocomposite is confirmed by NEXAFS spectroscopy data (Figure 7a).

It is significant that the relative carbon atoms amount in the C=O compound with respect to carbon atoms on the nanotube surface, determined by the ratio of the areas of the peaks A and B′ in the WC$_{1-x}$/MWCNT C1 spectrum (4.7%), is twice as large than the oxygen atoms proportion in the oxides adsorbed on the initial nanotube surface (2.3%). Since the peak B′ in the nanocomposite XPS C1 spectrum indicates the chemical interaction of the MWCNT carbon atoms with oxides initially located on its surface, i.e., the formation of a C=O bond at a temperature of 300 °C during the tungsten carbide deposition. Therefore, the observed carbon atom amount increase is possible with the participation of the portion of the carbon atoms on the MWCNT surface in the tungsten carbide formation during pyrolysis.

Figure 9b shows the results of the composite O1-line fitting with four components O (530.4 eV), B′ (531.2 eV), O′ (532.1 eV) and O″ (537.2 eV). When comparing the XPS O1 spectra of the initial nanotubes and the WC$_{1-x}$/MWCNT composite shown in Figure 9b, the additional intense O peak (530.4 eV) is observed. The energy position of this peak is in good agreement with the main peak in the WO$_3$ spectra [85,86]. This suggests that, in addition to the WC$_{1-x}$ bulk layer, a tungsten oxide layer forms on the composite surface. It can be seen from the figure that the peaks B′ (531.2 eV) and O′ (532.1 eV) are shifted toward lower energies relative to the peaks B and C positions in the nanotube spectrum by 0.4 eV and 1.3 eV, respectively. The observed "red" shift of the binding energies indicates a change in the state of oxygen atoms on the MWCNT surface and the formation of new compounds.

According to X-ray diffractometry, tungsten carbide coatings are nanosized crystallites of the nonstoichiometric WC$_{1-x}$ phase. Taking this into account, it is also reasonable to expect the tungsten oxide formation in the nonstoichiometric WO$_{3-x}$ phase. In [87], it was shown that the XPS O1 spectra of the nonstoichiometric crystalline WO$_{3-x}$ contain two peaks. The low-energy peak with the energy 530.5 eV corresponds to the O$^{2-}$ ion of WO$_3$, and the high-energy peak—532.6 eV, identical to the O1 peak of the O$^{2-}$ ion of WO$_{3-x}$ other combination states. It was shown in [85,88,89] that free

oxide surfaces contacting with the atmosphere are always hydrated, i.e., contain water molecules and hydroxyl groups. Component O′ and O″ located at 532.1 eV and 537.2 eV illustrates the water bound at the samples surface, proving the existence of $WO_3(H_2O)n$ phases at the surface. Therefore, the peaks O′ and O″ in the O1 spectrum of the nanocomposite should be identified as excitations in the tungsten oxide layer. In this case, the peak B′ energy position is in good agreement with the oxygen atom binding energy in the C=O group (531.6 eV) [90], which confirms the presence of an oxygen-carbon double bond in the nanocomposite and is consistent with the $WC_{1-x}$/MWCNT NEXAFS and XPS C1 spectra, indicating the C=O group presence. From the comparison of the peak B′ areas of the nanocomposite XPS C1 and O1 spectra, the ratio of carbon and oxygen atoms participating in the oxides formation on the nanotube surface was found to be 2.3. This means that carbon forms not only the C=O bond but also compounds with higher oxygen coordination, for example, such as the epoxy O–C–O bond and the $[CO_3]^{2-}$ anion. This is consistent with NEXAFS spectroscopy data (Figure 7a).

Figure 9c shows the W4f line deconvolution in five components $M_1$ (32.13 eV), $M_2$ (34.20 eV), $O_1$ (35.53 eV), $O_2$ (37.67) eV and $O_3$ (40.95 eV). The binding energies of the peaks $M_1$ and $M_2$ are close to the binding energies of the $W4f_{7/2,5/2}$ levels of crystalline WC (31.8 eV and 33.96 eV) [80]. According to [64,91], binding energies of 32.0 and 34.1 eV are associated with the presence of $WC_{1-x}$. The XRD analysis results are in good agreement with the fact that the non-stoichiometric $WC_{1-x}$ phase predominates in the coatings. M and $M_1$ bands in the carbon and tungsten spectra are responsible for the W–C chemical bond formation. A comparison of their integrated intensities makes it possible to determine the carbide stoichiometric formula as $WC_{0.88}$. The peaks $O_1$ and $O_2$ maxima correspond to $W4f_{7/2}$- and $W4f_{5/2}$- levels of oxide $WO_{3-x}$ tungsten atoms [85,86,92]. The O and $O_1$ bands in the oxygen and tungsten spectra are responsible for the W–O chemical bond formation. A comparison of their integrated intensities makes it possible to determine the oxide stoichiometric formula as $WO_{2.77}$. The peak $O_3$ maximum corresponds to $W5p_{3/2}$- levels of $W^{6+}$ ion in oxide $WO_3$ [93].

According to TEM data, the MWCNTs coated with pyrolytic tungsten carbide (Figure 4b) have the average thickness of about 20–30 nm. Using this value of the $WC_{1-x}$ layer thickness and the mean free paths of the photoelectrons emitted with certain energies from the C1 level in the tungsten carbide and oxide it is possible to determine the thickness of the $WO_{3-x}$ layer from the analysis of the nanocomposite XPS W4f spectra. The $I_l$ lines intensities in the XPS spectra describe the emission of electrons from the $l$ level, and have a complex dependence:

$$I_l = n_l \sigma_l J \lambda S \left( 1 - e^{-\frac{d}{\lambda \cos \theta}} \right), \tag{3}$$

where $n_l$ is the concentration of atoms, $\sigma_l$ is the cross section for the photoelectron emission from the atom $l$ subshell, which depends on the incident X-ray quantum energy $h\nu$, $J$ is the incident radiation intensity, $\theta$ is the photoelectrons emission angle measured from the normal to the sample, $\lambda$ is the electron mean free path in the studied sample, $S$ is the detector sensitivity. It should be noted that $\lambda$ is determined by the emitted photoelectron energy $E_{kin}$. S depends on the incident quantum energy $h\nu$, the specific of the analyzer operation, and shooting conditions. If the sample is semi-infinite, then the exponential term in formula (3) can be neglected.

For a two-layer structure, when sample B is covered with a thin layer of another substance $A$, the lines intensity in sample $B$ XPS spectrum can be written as:

$$I_B = n_B \sigma_B J_B \lambda_B S_B \left( 1 - e^{-\frac{d_B}{\lambda_B \cos \theta}} \right) e^{-\frac{d_A}{\lambda_A \cos \theta}}. \tag{4}$$

The second exponent in Formula (4) is related to the absorption of photoelectrons from sample B in the sample $A$ covering layer. If we consider XPS spectra taken in the same geometry and for one type of atom, but in different compounds, then the quantities $\sigma_l$, $J$, $S$ and $\theta$ will be about the same. In the present work, the detector was oriented normally to the sample surface, i.e., the angle $\theta = 0$. Taking the $WC_{1-x}$ layer thickness $d_C = 20$ nm from Equations (3) and (4), it is possible to determine $d_O$ (the

effective thickness of the $WO_{3-x}$ layer) from the ratios of the intensities of the peaks M and O in the XPS W4f spectra, which is equal to 3.3 nm. The photoelectrons mean free paths with an energy of 1445 eV in the oxide and in tungsten carbide were taken to be equal to the mean free paths for crystalline $WO_3$ ($\lambda = 1.89$ nm) and WC ($\lambda = 2.35$ nm) [94]. An analysis of the Equations (3) and (4) solution showed that at the available of these peaks intensities, a decrease in the carbide layer thickness up to $d_C = 7$ nm practically does not change the estimated tungsten oxide layer thickness $d_O$, i.e., the semi-infinite layer approximation is fair for $d_C > 7$ nm. Then, with a tungsten carbide layer thickness of 20–30 nm and $\lambda = 2.35$ nm, the tungsten carbide coating layer can be considered semi-infinite. This allows us to estimate the upper bound on the relative sizes of the nanotube open surface in the nanocomposite. If we assume that the coating has the same constant thickness, then in this case, the peak A area in the $W_{1-x}$/MWCNT nanocomposite C1 spectrum (Sample 2) will be proportional to the initial nanotube free surface. The ratio of the A peaks integrated intensities (areas) in the C1 spectra of the nanocomposite and MWCNTs is 0.17, which allows us to evaluate the minimum degree of the studied sample surface coverage with tungsten carbide at 83%.

The studies carried out suggest the following scenario for the tungsten carbide coatings formation on the nanotube surface. At the MOCVD deposition initial stage at 300 °C, an insignificant part of the nanotube outer layer carbon atoms interacts with oxides (OH, $H_2O$, $O_2$) adsorbed on the MWCNT surface to form carbon oxides C–O, C–O–C, C=O and $CO_3$. In this case, some of the carbon atoms on the surface enter into chemical bonds with tungsten atoms and an intermediate layer is formed between MWCNT and a coating of the non-stoichiometric tungsten carbide $WC_{0.88}$ nanoscale particles. Good adhesion of the pyrolytic tungsten carbide nanoparticles to the nanotube surface is ensured by the amount of the MWCNT carbon atoms taking part in the formation of a chemical bond with the $WC_{1-x}$ coating layer tungsten atoms. Upon subsequent transfer of the $WC_{1-x}$/MWCNT composite to the atmosphere, a layer of non-stoichiometric $WO_{2.77}$ 3.3 nm thick forms on the tungsten carbide surface as a result of water and oxygen molecule adsorption.

## 4. Conclusions

The method of ferrocene–toluene mixture pyrolysis in Ar flow at atmospheric pressure using a tubular quartz reactor proved to be an effective technology for the synthesis of the initial MWCNT. The graphene layers form the outer surface of the MWCNT. The distance between neighboring graphene layers is 0.34 nm. The internal diameter, external diameter and length of the MWCNTs are about 80 nm, 6–10 nm and 0.1–2 mm, respectively. The oxygen atom percentage (OH and $H_2O$ oxides) on the outer surface of the MWCNT using NEXAFS, XPS and Raman scattering methods was found to be 3%.

It was found that the tungsten hexacarbonyl [$W(CO)_6$] pyrolysis technique allows us to obtain nonstoichiometric $WC_{1-x}$ nanoparticles layers on the MWCNT surface. SEM, TEM, HRTEM, FFT, EDS and XRD studies show that a thick coating of pyrolytic tungsten carbide varies in the wide interval and is granular. All this suggests that coatings are not deposited as continuous thin layers covering the entire MWCNT surface, they become continuous only if grains increase sizes and coalesce at a relatively high initial concentration of $W(CO)_6$. The XPS, NEXAFS and Raman spectroscopy data indicated the oxidization of the tungsten carbide coatings in air and formation of the $WO_{3-x}$ coatings with a thickness of 3.3 nm. XPS and NEXAFS-spectroscopy results showed that the MWCNT surface was covered with tungsten carbide layer consisting of non-stoichiometric $WC_{1-x}$ nanoparticles. The MWCNT–(pyrolytic coating) interface structure was determined. It consists of three layers: a transition layer, in which MWCNT outer surface carbon atoms form bonds with coating layer tungsten atoms; a coating layer of $WC_{0.88}$ tungsten carbide; and an outer layer of $WO_{2.77}$ tungsten oxide. The nonstoichiometric $WC_{1-x}$ nanoparticle adhesion is provided by the chemical bonding between the carbon atoms of the MWCNT outer layer and the tungsten atoms of the coating. In the work, the MOCVD technology for the tungsten hexacarbonyl decomposition at a temperature of 300 °C was used, which allowed us to obtain the tungsten carbide coating with high adhesion to the MWCNT surface.

**Supplementary Materials:** The following are available online at http://www.mdpi.com/2076-3417/10/14/4736/s1, Figure S1: a schematic illustration of the experimental setup for growing of the radial-oriented aligned MWCNTs, Figure S2: a schematic illustration of the experimental setup for synthesis of the WC$_{1-x}$/MWCNTs nanocomposite, Figure S3: The processing stages scheme.

**Author Contributions:** Conceptualization, D.S.; research supervision, writing—original draft, D.S., S.N. and V.S.; Synthesis of the nanocomposite samples, XRD measurement and its data interpretation, A.O., B.K., A.A. and I.V.; TEM, SEM and EDS measurements, S.G.; Raman spectroscopy measurements, S.I.; XPS and NEXAFS spectroscopy measurements and their data analysis, D.S., A.V., O.P., A.M., P.M., V.S. and S.N. All of the authors discussed the results and commented on the manuscript during its preparation. All authors have read and agreed to the published version of the manuscript.

**Funding:** The reported study was funded by RFBR, project numbers 19-32-60018 and 18-33-00776, partially funded by RSF, project number 18-79-10227, the bilateral program RGBL at BESSY II, project number 2019-1-19107863, and the state research target for the G.A. Razuvaev Institute of Organometallic Chemistry of Russian Academy of Sciences (theme No. 45.8).

**Acknowledgments:** Scientific research were performed at the Center for Studies in Surface Science of Research park of St. Petersburg State University and at the Centre of Collective Use at the Institute of Geology of Komi Science Centre of the Ural Branch of RAS.

**Conflicts of Interest:** The authors declare no conflict of interest.

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
