# Peer review of "Studies of Buried Layers and Interfaces of Tungsten Carbide Coatings on the MWCNT Surface by XPS and NEXAFS Spectroscopy"

_applsci, doi:10.3390/app10144736_

Round 1

Reviewer 1 Report

In this work the authors have done a careful study of the interfacial organization between multi-wall carbon nanotubes (MWCNT) and pyrolytic coatings of tungsten carbide. After a delicate analysis, they propose a 3-layer interface organization with the participation of oxide forms. This works is serious and deserves certainly to be published. Nevertheless, I have some concerns which will need to be solved before publication as listed below:

  • The paper organization is not appropriate. In fact, the work is focused in the determination of the structure of an interfacial structure between MWCNTs and CW coatings, but more than half of the abstract of the manuscript is devoted to exposing the interest of XPS and NEXAFS measurements to study buried interfaces. This type of information is not relevant here or could appear in a more reduced version AFTER presenting the obtained results in the abstract.

  • The same problem appears in the introduction part of the paper. The focus should be first placed on the material under study. XPS and NEXAFS are very powerful techniques extremely appropriate for the study that the authors have conveyed, but they are also very well documented techniques. Details should appear in the “Materials and Methods” section of the paper.

  • The authors refer in various parts of their manuscript to the good adhesion of the nanoparticles. Once there is a closed layer of the coating surrounding the nanotube we can certainly be sure that the integrity of the system is better preserved, but do the authors can provided experimental evidence of the good adhesion of the nanoparticles in the first growing stages as those shown in Fig. 4 ? The authors should provide some evidence that not only van der Waals forces apply in this case, for instance by observing the evolution of the sample after washing or other solvent cleaning. This is an important point as this good adhesion is evoked to construct the author interpretation of the NEXAFS and XPS data.

  • As seen in Fig. 3.c, the internal cross-section of a broken MWCNT appearing on the top part of the image is not totally circular. Coating can introduce some strain on the MWCNT and be responsible of cross-section modification as it has been seen in the case of MWCNT-polyamide composites submitted to high pressure [F. Balima et al, Carbon 106 (2016)]. This needs to be mentioned.

  • In Fig 3 caption as well as in subsequent figure captions, it should be indicated to which sample (sample 1, 2 or 3) corresponds the figure or figure panels.

  • There are a few typos. I’ve detected the following (in capitals the word corrected, changed or missed):

Line 56. It should read “In THIS work, …”

Line 126. It should read “…ratios were less THAN 3, …”

Line 216. Replace “equals” by “equal”

Line 552. Replace “The studies carried out to suggest…” by “The studies carried out  suggest…”

  • Line 175. I think it will be more appropriate to refer to the iron nanoparticles as “residual iron nanoparticles”

Author Response

The authors are grateful to the reviewer for attention to the work and valuable comments and corrections. Our responses to the comments are given below.

  • The paper organization is not appropriate. In fact, the work is focused in the determination of the structure of an interfacial structure between MWCNTs and CW coatings, but more than half of the abstract of the manuscript is devoted to exposing the interest of XPS and NEXAFS measurements to study buried interfaces. This type of information is not relevant here or could appear in a more reduced version AFTER presenting the obtained results in the abstract.

Response: Such paper organization was chosen in accordance with the theme of the special issue “Applications of X-ray Photoelectron Spectroscopy (XPS)”, therefore, the XPS method and its complementary NEXAFS method have been given a lot of space in the abstract.

  • The same problem appears in the introduction part of the paper. The focus should be first placed on the material under study. XPS and NEXAFS are very powerful techniques extremely appropriate for the study that the authors have conveyed, but they are also very well documented techniques. Details should appear in the “Materials and Methods” section of the paper.

Response: We do not agree with this remark. In the introduction, a large part is devoted specifically to the problem of WC/MWCNT nanocomposites preparation and description of its properties and methods of study. Only 15% of the text at the beginning of the introduction is devoted to XPS and NEXAFS methods.

The authors agree that the details of these methods should be presented in the “Materials and Methods” section, so we add in section 2.3 the text describing the features of the XPS and NEXAFS methods used in this work. 

  • The authors refer in various parts of their manuscript to the good adhesion of the nanoparticles. Once there is a closed layer of the coating surrounding the nanotube we can certainly be sure that the integrity of the system is better preserved, but do the authors can provided experimental evidence of the good adhesion of the nanoparticles in the first growing stages as those shown in Fig. 4 ? The authors should provide some evidence that not only van der Waals forces apply in this case, for instance by observing the evolution of the sample after washing or other solvent cleaning. This is an important point as this good adhesion is evoked to construct the author interpretation of the NEXAFS and XPS data.

Response:In the present work, the presence of bonds between the carbon atoms of the nanotube outer surface and the tungsten and oxygen atoms of the coating layer demonstrated through the analysis of data obtained by NEXAFS and XPS methods. Thus, it is these methods that are used to explain the adhesion of the layer to the surface of the MWCNT. The Fig. 4 only demonstrates that WC1-x particles sizes are of the order of several of tens nanometers.

  • As seen in Fig. 3.c, the internal cross-section of a broken MWCNT appearing on the top part of the image is not totally circular. Coating can introduce some strain on the MWCNT and be responsible of cross-section modification as it has been seen in the case of MWCNT-polyamide composites submitted to high pressure [F. Balima et al, Carbon 106 (2016)]. This needs to be mentioned.

Response: We have carefully studied the work [F. Balima et al., Carbon 106 (2016)] which investigated the structural evolution of multi-walled carbon nanotubes in a polyamide matrix as a function of applied high pressure up to 5GPa. This is interesting information for us since we conducted research on nanocomposites based on MWCNT after barothermic treatment in argon. We believe that this work is not relevant to our case. The asymmetry observed in Fig. 3 is an optical effect due to the uneven break edges of the coating layer and the nanotube itself.

  • In Fig 3 caption as well as in subsequent figure captions, it should be indicated to which sample (sample 1, 2 or 3) corresponds the figure or figure panels.

Response: We have added the designation of the samples in the captions to the Fig. 3, 4, 8, 9.

  • There are a few typos. I’ve detected the following (in capitals the word corrected, changed or missed):

Line 56. It should read “In THIS work, …”

Line 126. It should read “…ratios were less THAN 3, …”

Line 216. Replace “equals” by “equal”

Line 552. Replace “The studies carried out to suggest…” by “The studies carried out  suggest…”

  • Line 175. I think it will be more appropriate to refer to the iron nanoparticles as “residual iron nanoparticles”

Response: All of these typos have been corrected in the manuscript.

Reviewer 2 Report

This paper presented an interesting investigation of the pyrolytic tungsten carbide nanoscale coatings on multi-walled carbon nanotubes (MWCNTs) surfaces. The authors first reviewed the state of the art in this field. Then the experimental procedures were presented in detail. The authors later discussed multiple experimental results collected using scanning and transmission electron microscopy, X-ray diffractometry, Raman scattering, and NEXAFS spectroscopy. The scientific discovery of this paper is interesting to potential readers. The reviewer suggests several minor revisions and clarifications before publishing this paper. 

  1. The repeatability of the reported experimental results was not discussed. How many samples were tested in each type of experiment and how repeatable were these results? Please clarify in the revised paper. 
  2. Subtitles of 3.4 and 3.5 can be revised. Although the results reported in these two sections are NEXAFS Spectroscopy and XPS, just call them certain research may not be clearly enough to the readers. 

Author Response

The authors are grateful to the reviewer for attention to the work and valuable comments. Our responses to the comments are given below.

Point 1: The repeatability of the reported experimental results was not discussed. How many samples were tested in each type of experiment and how repeatable were these results? Please clarify in the revised paper.

Response: NEXAFS and XPS measurements were carried out on powders rubbed into a metal holder. The measurements were carried out for a series of samples at different points on the surface of each sample. We have added the correspond information to the section 2.3.

Point 2: Subtitles of 3.4 and 3.5 can be revised. Although the results reported in these two sections are NEXAFS Spectroscopy and XPS, just call them certain research may not be clearly enough to the readers. 

Response: Subtitles of 3.4 and 3.5 revised to“3.4. NEXAFS Spectroscopy Research of the Nanocomposites and Initial MWCNTs”, and “3.5. XPS Research of the Nanocomposites and Initial MWCNTs”